# Risk of Gynecological Cancers in Cholecystectomized Women: A Large Nationwide Cohort Study

**DOI:** 10.3390/cancers14061484

**Published:** 2022-03-14

**Authors:** Elham Kharazmi, Kristina Sundquist, Jan Sundquist, Mahdi Fallah, Justo Lorenzo Bermejo

**Affiliations:** 1Institute of Medical Biometry, University of Heidelberg, 69120 Heidelberg, Germany; kharazmi@imbi.uni-heidelberg.de; 2Risk Adapted Prevention Group, Division of Preventive Oncology, German Cancer Research Center (DKFZ), 69120 Heidelberg, Germany; m.fallah@dkfz.de; 3Center for Primary Health Care Research, Lund University, 202 13 Malmö, Sweden; kristina.sundquist@med.lu.se (K.S.); jan.sundquist@med.lu.se (J.S.); 4Department of Family Medicine and Community Health, Icahn School of Medicine at Mount Sinai, New York, NY 10029, USA; 5Center for Community-Based Healthcare Research and Education (CoHRE), Department of Functional Pathology, School of Medicine, Shimane University, Izumo 693-8501, Japan; 6Institute of Primary Health Care (BIHAM), University of Bern, 3012 Bern, Switzerland

**Keywords:** gallbladder surgery, cholecystectomy, breast cancer, endometrial cancer, ovarian cancer, cervical cancer, large nationwide cohort

## Abstract

**Simple Summary:**

Gallstones affect women more frequently than men. Symptomatic gallstones are treated with surgical removal of the gallbladder. Overweight, obesity, and exposure to female hormones can cause gallstones and also breast, uterine, and ovarian cancer. We assessed if surgical removal of the gallbladder is associated with these cancers in women. We found risk of ovarian cancer is increased within the first 6 months after surgery. Women undergoing this operation also show an increased risk of breast and uterine cancer up to 30 years after surgery. It is important to screen women with this surgery indication for the three abovementioned cancers.

**Abstract:**

**Background**: Gallstones affect women more frequently than men, and symptomatic gallstones are increasingly treated with surgical removal of the gallbladder (cholecystectomy). Breast, endometrial, and ovarian cancer share several risk factors with gallstones, including overweight, obesity, and exposure to female sex hormones. We intended to assess the association between cholecystectomy and female cancer risk, which has not been comprehensively investigated. **Methods:** We investigated the risk of female cancers after cholecystectomy leveraging the Swedish Cancer, Population, Patient, and Death registries. Standardized incidence ratios (SIRs) adjusted for age, calendar period, socioeconomic status, and residential area were used to compare cancer risk in cholecystectomized and non-cholecystectomized women. **Results:** During a median follow-up of 11 years, 325,106 cholecystectomized women developed 10,431 primary breast, 2888 endometrial, 1577 ovarian, and 705 cervical cancers. The risk of ovarian cancer was increased by 35% (95% confidence interval (CI) 2% to 77%) in the first 6 months after cholecystectomy. The exclusion of cancers diagnosed in the first 6 months still resulted in an increased risk of endometrial (19%, 95%CI 14% to 23%) and breast (5%, 95%CI 3% to 7%) cancer, especially in women cholecystectomized after age 50 years. By contrast, cholecystectomized women showed decreased risks of cervical (−13%, 95%CI −20% to −7%) and ovarian (−6%, 95%CI −10% to −1%) cancer. **Conclusions:** The risk of ovarian cancer increased by 35% in a just short period of time (6 months) following the surgery. Therefore, it is worth ruling out ovarian cancer before cholecystectomy. Women undergoing cholecystectomy showed an increased risk of breast and endometrial cancer up to 30 years after surgery. Further evaluation of the association between gallstones or gallbladder removal on female cancer risk would allow for the assessment of the need to intensify cancer screening in cholecystectomized women.

## 1. Introduction

Surgical removal of the gallbladder (cholecystectomy) is indicated mainly for the treatment of symptomatic gallstones (cholelithiasis) and its complications. Considering the high and increasing prevalence of gallstones, cholecystectomy is one of the most commonly performed surgeries [1,2]. Several risk factors for gallstone formation have been identified. One of the most important risk factors is female gender; depending on age, women have a two to three times higher risk of developing gallstones than men [3,4]. Pregnancy, which is characterized by a physiological increase in estrogen and progesterone, is a major risk factor for gallstone formation in women, and the risk of gallstone disease also increases with increasing parity [5,6]. Hormone replacement therapy in postmenopausal women and the use of oral contraceptives are also associated with an increased risk of cholelithiasis [7,8,9]. Both obesity and rapid weight loss are established risk factors for gallstone formation [10,11,12].

Gallstone disease and breast, endometrial, and ovarian cancers share several important risk factors, including overweight, obesity, and exposure to female sex hormones [13,14]. Cervical cancer and gallstones are both associated with low socioeconomic status, use of oral contraceptives, and parity [15,16,17,18,19]. Despite the shared risk factors and some reports that cholecystectomy is associated with an increased risk of endometrial and breast cancer, the association between cholecystectomy and female cancer risk has not been comprehensively investigated.

In the present large nationwide cohort study, we ascertained and accurately quantified the association between cholecystectomy and the risk of breast, endometrial, ovarian, and cervical cancer. We also considered the time since cholecystectomy, and the age at cholecystectomy for the identified associations. Our aim was to find new clues for the early detection and improved prevention of the investigated female cancers.

## 2. Materials and Methods

### 2.1. Study Datasets

For the current study, data from the Swedish National Patient Registers, the Swedish Cancer Registry, national censuses, and the Death Register were linked using the individually unique pseudonymized national registration number. These nationwide registered sources are updated every 1–3 years, and in our datasets, which were updated in 2017, the combined datasets included information for more than 15 million individuals up to the end of 2015 (there is a 2-year delay in the release of cancer registry data to ensure the completeness of the reported cases). The Swedish National Patient Registers include nationwide data on surgical procedures from all private and public hospitals and visits to specialized physicians in Sweden, as well as hospital (inpatient) records from 1964 to 2015 and day clinic (outpatient) records from 2001 to 2015. Information on cholecystectomy (using International Classifications of Diseases (ICD) code, 9th revision (ICD-9] 51.2 and its subcategories) and date of the surgery were extracted from this dataset. The Swedish Cancer Registry data provided information on date of cancer diagnosis, topography and morphology, and diagnostic reports from physicians for 1958–2015. All cancer records were reported using ICD-7. There were no missing data on the cancer status of individuals. The overall completeness of the registry was estimated to be 96% (for non-hematological organs, including our cancers of interest, it is probably even higher) [20]. Demographic information, such as sex, birth, migration data, and information on date of death were obtained from the National Population Register, the national censuses, and the Cause-of-Death Register.

### 2.2. Follow-Up

The final dataset contained up to 51 years of follow-up. The follow-up for patients with cholecystectomy started at the date of cholecystectomy (month and year) and for other individuals it was from the beginning of 1964, the birth date (month and year), or the immigration, whichever was the latest. The follow-up period for all individuals ended when they were diagnosed with cancer, emigrated, died, or at the end of study period (end of 2015), whichever came first.

### 2.3. Statistical Analyses

Standardized incidence ratios (SIRs) were calculated to compare the risks of first primary breast, endometrial, ovarian, and cervical cancer diagnoses for women with a history of cholecystectomy with the risks in their counterparts without such a history. SIRs were calculated as the ratio of observed to expected number of incident cancer cases. The expected numbers were calculated from the strata-specific cancer incidence rate in those with no history of cholecystectomy, multiplied by the corresponding person-years for subjects with history of cholecystectomy. All SIRs were adjusted for age, calendar period (5-year interval), socioeconomic status (farmer, manual workers, low to middle income office worker, high income office worker/professional, and company owner except farmer or other/unspecified), and residential area (large cities, South Sweden, North Sweden, or unspecified). Confidence intervals (95% CI) were calculated assuming a Poisson distribution. Further stratified analyses were performed by age at cholecystectomy (<40, 40–49, ≥50 years) and time since cholecystectomy (1–6, 7–12, 13–36, and ≥37 months). *P*-values were adjusted for multiplicity using the Bonferroni method (0.05 divided by four investigated cancer types = 0.0125; *p*-values smaller than 0.0125 are highlighted with an asterisk in the tables and figures). For cholecystectomized women, cancer incidence rates by year since surgery were also calculated. Incidence rates in non-cholecystectomized women were calculated for each 1-year interval age and shown in the same chart starting from the corresponding median age of surgery in cholecystectomized women.

All analyses were performed using SAS software, version 9.4 (by SAS Institute Inc., Cary, NC, USA). To avoid the risk of identification of participants, researchers only had access to pseudonymized secondary data. All linkages were performed by the individual national identification number, which is assigned to all residents staying in Sweden longer than 3 months (residence permit). To preserve confidentiality, this ID number was replaced by a serial number.

## 3. Results

Among the 325,106 cholecystectomized women, 10,431 primary breast, 2888 endometrial, 1577 ovarian, and 705 cervical cancers were diagnosed during a median follow-up of 11 years. Considering any time elapsed since gallbladder removal, cholecystectomized women showed an increased risk of endometrial (18%, 95% CI 14% to 23%) and breast cancer (4%, 95% CI 2% to 6%; Table 1, column “All”).

The proportion of cancers that theoretically could potentially be attributable to cholecystectomy in the general population was at most 0.2% for breast cancer and 1% for endometrial cancer. In those with both a history of cholecystectomy and a diagnosis of gynecological cancers, the proportion of cholecystectomies performed after the cancer diagnoses in our dataset was 29.9% for breast cancer, 62.7% for cervical cancer, 33.1% for endometrial cancer, and 32.3% after ovarian cancer.

### 3.1. SIRs by Time after Cholecystectomy

The risk of ovarian cancer increased by 35% (95% CI 2% to 77%) in the first 6 months after cholecystectomy (95% CI did not include 1), but the risk increase did not reach statistical significance after multiplicity adjustment (*p*-value = 0.03, Table 1). Three years after surgery, cholecystectomized women showed an increased risk of endometrial (18%, 95% CI 14% to 23%) and breast cancer (4%, 95% CI 2% to 6%, Table 1 and Figure 1). Women with cholecystectomy showed a lower risk of cervical cancer from 3 years after the surgery (−13%, 95% CI −20% to −6%).

When cancer diagnoses within the first 6 months after gallbladder removal were excluded, cholecystectomized women showed an increased risk of endometrial (19%, 95% CI 14% to 23%) and breast cancer (5%, 95% CI 3% to 7%; Table 2, column “All”). By contrast, women with cholecystectomy showed lower risks of cervical (−13%, 95% CI −20% to −7%) and ovarian cancer (−6%, 95% CI −10% to −1%).

### 3.2. SIRs by Age at Cholecystectomy

When only cancers diagnosed at least 7 months after cholecystectomy were considered, the risk of breast cancer slightly increased with increasing age at cholecystectomy (from no increased risk for age at cholecystectomy under 40 years, to 7% risk increase for surgery after age 49 years; Table 2). The risk of endometrial cancer in women cholecystectomized before age 40 years was increased by 18%, whereas women with gallbladder surgery after age 49 years showed a slightly higher risk increase (20%), however with overlapping 95% confidence intervals.

### 3.3. Age-Specific Incidence Curves

The graphical comparison of the age-specific incidence curves for cancers with an increased risk in cholecystectomized women revealed that the risk of breast cancer in women with a gallbladder surgery before 40 years old was similar to the risk of breast cancer in non-cholecystectomized women (Figure 2). By contrast, the risk of breast cancer after cholecystectomy at age 40–49 years increased from 7 years after surgery, and the risk of breast cancer after cholecystectomy at age ≥50 years was increased between 7 and 22 years after surgery (Figure 2). 

The risk of endometrial cancer after cholecystectomy was higher than in non-cholecystectomized women (Figure 3), and the general tendency was an increasing risk difference with increasing time after surgery for at least three decades.

## 4. Discussion

In this large nationwide cohort study with a follow-up of up to 51 years (median 11 years), we were able to quantify the risk of breast and gynecological cancer after cholecystectomy in women by time since surgery, and age at surgery, with unprecedented precision. Overall, a 15% to 20% increased risk of endometrial cancer, and a 4% to 8% increased risk of breast cancer after cholecystectomy were observed. Ovarian cancer showed a 35% increased risk during the first half-year period after cholecystectomy, but the risk of ovarian cancer did not increase after 3 years from gallbladder removal. Cholecystectomies performed after the age of 50 years were associated with a lower risk of cervical cancer.

Dietary habits (especially high intake of fats and refined carbohydrates), obesity, and the associated hormonal and metabolic changes may explain part of the observed association between cholecystectomy (most often indicated because of cholelithiasis) and breast cancer (especially post-menopausal type), endometrial cancer, and ovarian cancer [21,22,23,24]. The increased risk of ovarian cancer found in the first 6 six months after cholecystectomy could be attributed in part to misdiagnosing the reason for pain in the right upper quadrant of the abdomen as pain caused by gallstones and performing cholecystectomy, even though the pain was actually related to ovarian cancer. A modeling study has also recently found that the opportunistic salpingectomy performed at the time of elective laparoscopic cholecystectomy may be a cost-effective strategy to prevent ovarian cancer among average-risk women [25].

Surgical removal of the gallbladder can lead to changes in metabolic hormone levels and bacterial microbiota, which in turn can cause inflammation—this sequence has been hypothesized as a likely reason for the increased risk of cancer after cholecystectomy [26,27,28,29,30,31,32,33,34,35,36,37,38]. The finding that, in women with both history of cholecystectomy and gynecological cancers, about 30–63% of cholecystectomies were performed after the diagnosis of gynecological cancers further supports the idea that common hormonal or metabolic causes underneath both cholecystectomy and gynecological cancers and not the surgical removal of gallbladder per se. Our findings suggest that the increased risk of breast and endometrial cancer after cholecystectomy is probably due to common risk factors (such as diet, obesity, and associated metabolic and hormonal changes) that result in the cholelithiasis and therefore are an indication for cholecystectomy and at the same time increasing risk of breast and endometrial cancer.

The increased risk of ovarian cancer detected in the first half-year period after cholecystectomy could be related to existing undiagnosed ovarian cancer. Therefore, women with an indication for cholecystectomy, especially those with overweight or obesity, should be screened for ovarian, endometrial, and breast cancer, and should also take steps to improve their dietary habits, weight control, and exposure to exogenous estrogen after gallbladder surgery to avoid these cancers. Further studies on the exact mechanisms behind the identified associations and age/type/frequency of intensified cancer screening procedures are warranted, which may guide future cancer prevention measures.

When cholecystectomy was performed at the age of 50 years or older, it was associated with a decreased risk of cervical cancer, whereas cholecystectomy at younger ages was not associated with a lower risk of cervical cancer. This observation needs to be confirmed in future studies, and the reason underlying the observed decrease in risk needs to be further clarified. Cervical cancer shares some risk factors with gallstone disease, such as low socioeconomic status, oral contraceptive use, and parity. However, the main cause of cervical cancer is human papilloma virus infection [39].

Information on the subtypes of cancers (e.g., triple negative breast cancer) and the therapeutic response of patients were not available for our study data. Further studies are needed to find the association between surgical gallbladder removal and cancer stage, disease progression, and cancer sub-types, e.g., triple negative breast cancer, high grade serous ovarian carcinoma, and clear cell carcinoma. It is also warranted to find any differences in the survival or therapeutic response of the patients who had undergone cholecystectomy. It would be worth investigating in further studies whether such associations are significantly different in the presence or absence of a family history of the corresponding gynecological cancer.

According to the results of our comprehensive analyses of a large nationwide cohort, which was based on highly accredited long-lasting register datasets, we provided accurate risk estimates according to the time since cholecystectomy and the age at cholecystectomy. We investigated the risk patterns considering these two important predictive factors, and reported novel, clinically relevant information that was lacking in previous studies.

Our study benefitted from long-standing and large-scale nationwide data on diseases and surgical procedures in Sweden since 1964, with the possibility of record linkage to a high-quality nationwide cancer registry that was in operation since 1958 with a completeness of over 96%. This allowed us to comprehensively analyze the cancer risk after cholecystectomy by time since cholecystectomy and age at surgery with unprecedented detail. The surveillance bias was mitigated by the exclusion of cancer diagnoses within first 6 months after cholecystectomy.

In recent years, numerous studies have reported that the detection rate of gynecological cancers, except ovarian cancer, has been improving, especially for those with existing recommended screening modalities. Therefore, in our long-term study (1964–2015), in addition to aging, the effect of the year of inspection on the detection rate of breast, endometrial, ovarian, and cervical cancer could have partially contributed to the overall increasing trend of incidence of these cancers by time after cholecystectomy in both cholecystectomized and reference groups. However, without using such a long-standing data, the long-term follow-up of cholecystectomy would not be possible. There were some random fluctuations at the end of the incidence curves, which were mainly due to small sample size at elderly ages (25–30 years after cholecystectomy). The chance of findings due to multiple testing was restricted by considering probability values smaller than 0.0125 as statistically significant. Similar to other register-based studies, detailed clinical information, such as the exact indication for cholecystectomy, was lacking, although the majority of patients globally undergo this surgery due to symptomatic cholelithiasis or its complications. The residual confounding factors, such as obesity, diet, ethnicity, cigarette smoking, education, and physical activity, may co-exist. We adjusted all SIRs for residential area and socioeconomic status that, to some extent, could take care of variation in lifestyle factors.

In conclusion, the risk of ovarian cancer increased by 35% in just a short period of time (6 months) following the surgery. Therefore, it is worth ruling out ovarian cancer before cholecystectomy. Women undergoing cholecystectomy also showed an increased risk of breast and endometrial cancer up to 30 years after surgery. Further evaluation of the association between gallstones or gallbladder removal on female cancer risk would allow for the assessment of the need to intensify cancer screening in cholecystectomized women. In other words, women with an indication for cholecystectomy could benefit from screening for these cancers.

## Figures and Tables

**Figure 1 cancers-14-01484-f001:**
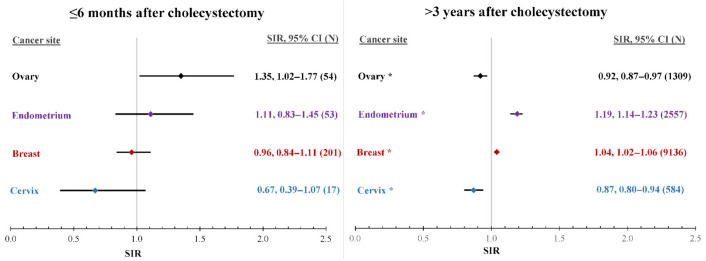
Relative risk of female cancers after cholecystectomy. Left panel: During the first six months after surgery. Right panel: Diagnosed more than three years after cholecystectomy. *: Probability value smaller than 0.0125 (0.05 divided by the four investigated female cancer sites); N: number of cancer patients; SIR: standardized incidence ratio; CI: confidence interval.

**Figure 2 cancers-14-01484-f002:**
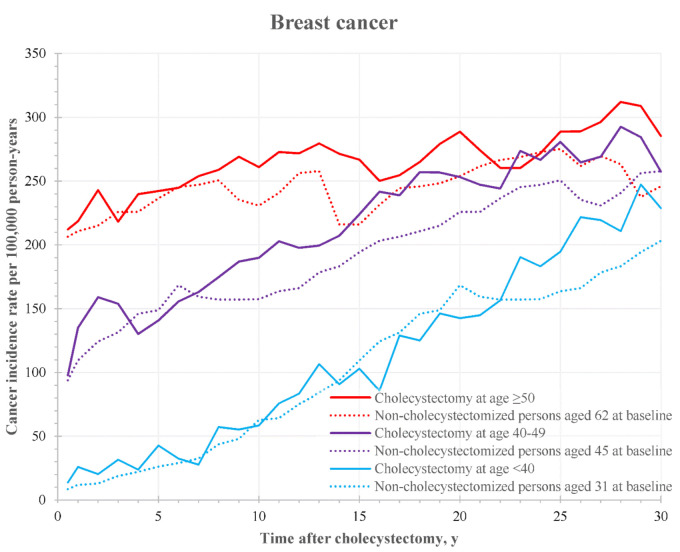
Incidence rate of breast cancer after cholecystectomy by time after surgery and patient’s age at the operation. Dotted lines represent the corresponding rates for non-cholecystectomized individuals with an age at baseline equal to the median age in the groups of cholecystectomized patients.

**Figure 3 cancers-14-01484-f003:**
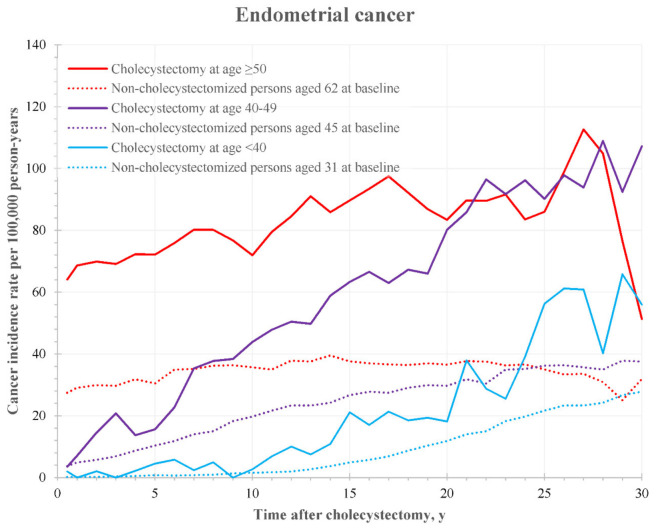
Incidence rate of endometrial cancer after cholecystectomy by time after surgery and patient’s age at the operation. Dotted lines represent the corresponding rates for non-cholecystectomized individuals with an age at baseline equal to the median age in the groups of cholecystectomized patients.

**Table 1 cancers-14-01484-t001:** Relative risk of female cancers by time after cholecystectomy.

	Time after Cholecystectomy																				
	1–6 Months				7–12 Months				13–36 Months				>3 Years				All				
Cancer	N	SIR	95% CI		N	SIR	95% CI		N	SIR	95% CI		N	SIR	95% CI		N	SIR	95% CI		ICD-7
**Breast**	201	0.96	0.84	1.11	217	1.05	0.92	1.20	877	**1.08**	1.01	1.15	9136	**1.04 ***	1.02	1.06	10,431	**1.04 ***	1.02	1.06	170
**Cervix**	17	0.67	0.39	1.07	16	0.65	0.37	1.05	88	0.94	0.75	1.15	584	**0.87 ***	0.80	0.94	705	**0.86 ***	0.80	0.93	171
**Endometrium**	53	1.11	0.83	1.45	55	1.16	0.88	1.51	223	**1.19 ***	1.04	1.36	2557	**1.19 ***	1.14	1.23	2888	**1.18 ***	1.14	1.23	172
**Ovary**	54	**1.35**	1.02	1.77	45	1.14	0.84	1.53	169	1.10	0.94	1.28	1309	**0.92 ***	0.87	0.97	1577	0.95	0.91	1.00	175

N: number of primary cancers after cholecystectomy; SIR: incidence ratio standardized for age, calendar period, socioeconomic status, and residential area; CI: confidence interval; ICD-7: International Classifications of Diseases, 7th revision. Bold values indicate that the 95% CI did not include 1.00. *: Probability value smaller than 0.0125 (0.05 divided by the four investigated female cancer sites).

**Table 2 cancers-14-01484-t002:** Relative risk of female cancers by age at cholecystectomy. Cancers diagnosed in the first 6 months after cholecystectomy are excluded.

	Age at Cholecystectomy (Excluding Cancers Diagnosed in the First 6 Months after Cholecystectomy)															
	<40 Years				40–49 Years				≥50 Years				All			
Cancer	N	SIR	95% CI		N	SIR	95% CI		N	SIR	95% CI		N	SIR	95% CI	
**Breast**	2583	1.00	0.96	1.04	2184	**1.05**	1.01	1.09	5463	**1.07 ***	1.04	1.10	10,230	**1.05 ***	1.03	1.07
**Cervix**	294	1.00	0.89	1.12	125	**0.82**	0.68	0.98	269	**0.78 ***	0.69	0.87	688	**0.87 ***	0.80	0.93
**Endometrium**	527	**1.18 ***	1.08	1.28	577	**1.15 ***	1.06	1.25	1731	**1.20 ***	1.14	1.26	2835	**1.19 ***	1.14	1.23
**Ovary**	362	0.99	0.89	1.10	306	**0.89**	0.79	1.00	855	0.94	0.88	1.01	1523	**0.94 ***	0.90	0.99

N: number of first cancers after cholecystectomy; SIR: standardized incidence ratio adjusted for age, calendar period, socioeconomic status and residential area; CI: confidence interval. Bold values indicate that the 95% CI did not include 1.00. *: Probability value smaller than 0.0125 (0.05 divided by the four investigated female cancer sites).

## Data Availability

This study leveraged the Swedish Cancer Registry and the Swedish National Patient Register. Raw data from these registers cannot be shared by the study authors, however further information and relevant contact details can be found on: https://www.socialstyrelsen.se/en/statistics-and-data/registers/ (last accessed on 12 March 2022) Links and email addresses for registers (last accessed on 12 March 2022): https://www.socialstyrelsen.se/en/statistics-and-data/registers/national-cancer-register/ (last accessed on 12 March 2022) (cancerregistret@socialstyrelsen.se) https://www.socialstyrelsen.se/en/statistics-and-data/registers/national-patient-register/ (patientregistret@socialstyrelsen.se) (last accessed on 12 March 2022).

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
