# Peer review of "Risk of Gynecological Cancers in Cholecystectomized Women: A Large Nationwide Cohort Study"

_cancers, 2022, doi:10.3390/cancers14061484_

Round 1

Reviewer 1 Report

The manuscript entitled: Risk of Gynecological Cancers in Cholecystectomized Women: A Large Nationwide Cohort Study, is based on a Sweden population dataset including more than 15 million people with up to 51 years follow-up. The follow-up period for all individuals ended when they either were diagnosed with cancer, emigrated, died, or at the end of study period, whichever came first. Cancer diagnosis data were complete (96% or higher) for the period 1958-2015. Records including surgical procedures and visits, both at hospitals (inpatients) and day clinic (outpatient), were complete from 2001 to 2015. Ethical issues. The study was approved by the Lund University ethics committee. Written consent is not required for the register-based studies in Sweden. However, through advertisements in the major newspapers, people could choose to opt out before the project database was constructed. Data confidentiality relied on replacing the national ID number by a serial number.

The authors aim was to find new clues for early detection and improved prevention of the investigated female cancers. According to the statistical analyses description, the work was properly done. Anyway, if possible, to follow open data policies, raw (grouped) data tables should be also published.

Here below are my specific comments to the authors.

  1. If there are common hormonal or metabolic causes underneath both cholecystectomy and gynecological cancers, although gallstones formation is expected to be a quicker process than cancer development, I would still expect to find some gynecological cancer patients who suffer from cholelithiasis and may even need a cholecystectomy after cancer diagnosis. If in 15 million people, there is not a single case of cholelithiasis and/or cholecystomy after gynecological cancer diagnoses, this point should be highlighted and discussed rather than assumed as obvious. This point is not important in terms of your objective (improving early cancer diagnoses) but could be relevant regarding the investigation of the underlying links or causes of cholelithiasis and gynecological cancers.
  2. I wonder whether the observed increase in certain gynecological cancers is associated to gallstones formation, to gall absence or to the surgical procedure. Were there no patients who had gallstones and were managed without undergoing a cholecystectomy? If such patients existed, did those patients develop gynecological cancers later or not? The association is the same if an open surgery or a laparoscopic surgery is done?
  3. The risk of ovarian cancer increased by 35% just in a small 6 months window following the surgery. I think that the interpretation that you proposed, namely a possible pain-based misdiagnosis, is relevant enough to be included in the abstract, highlighting that it is worth doing efforts to exclude ovarian cancer before cholecystectomy.
  4. Other works have explored a link among cholelithiasis or cholecystectomy and colon cancer, which is also very interesting. I wonder if you have explored or will explore this link in your large complete dataset of Sweden patients.

Author Response

The authors aim was to find new clues for early detection and improved prevention of the investigated female cancers. According to the statistical analyses description, the work was properly done. Anyway, if possible, to follow open data policies, raw (grouped) data tables should be also published.

  • Thank you very much for your comment. Study authors cannot publish nationwide raw data; however, we added the following section (page 8, last paragraph before References) to facilitate contact of readers to source data keepers:
    • Data Availability: This study leveraged the Swedish Cancer Registry as well as the Swedish National Patient Register. Raw data from these registers cannot be shared by study authors, however further information and relevant contact details can be found on: https://www.socialstyrelsen.se/en/statistics-and-data/registers/register-information/ Links and email addresses for registers: https://www.socialstyrelsen.se/en/statistics-and-data/registers/register-information/swedish-cancer-register/ ([email protected]) https://www.socialstyrelsen.se/en/statistics-and-data/registers/register-information/the-national-patient-register/ ([email protected]).

Here below are my specific comments to the authors.

  1. If there are common hormonal or metabolic causes underneath both cholecystectomy and gynecological cancers, although gallstones formation is expected to be a quicker process than cancer development, I would still expect to find some gynecological cancer patients who suffer from cholelithiasis and may even need a cholecystectomy after cancer diagnosis. If in 15 million people, there is not a single case of cholelithiasis and/or cholecystomy after gynecological cancer diagnoses, this point should be highlighted and discussed rather than assumed as obvious. This point is not important in terms of your objective (improving early cancer diagnoses) but could be relevant regarding the investigation of the underlying links or causes of cholelithiasis and gynecological cancers.

  • Thank you for your comment. We had no access to data on cholelithiasis in the Swedish nationwide study datasets to which we have access. However, we checked number of cholecystectomies after gynecological cancer diagnoses and added following sentence to Results and Discussion sections:
    • Results, page 4, first paragraph: In those with both history of cholecystectomy and diagnosis of gynecological cancers, proportion of cholecystectomies performed after the cancer diagnoses in our dataset was 29.9% for breast cancer, 62.7% for cervical cancer, 33.1% for endometrial cancer, and 32.3% after ovarian cancer.
    • Discussion, page 7, second paragraph: The finding that, in women with both history of cholecystectomy and gynecological cancers, about 30%-63% of cholecystectomies had been performed after diagnosis of gynecological cancers further supports the idea that common hormonal or metabolic causes underneath both cholecystectomy and gynecological cancers and not the surgical removal of gallbladder per se.

  1. I wonder whether the observed increase in certain gynecological cancers is associated to gallstones formation, to gall absence or to the surgical procedure. Were there no patients who had gallstones and were managed without undergoing a cholecystectomy? If such patients existed, did those patients develop gynecological cancers later or not? The association is the same if an open surgery or a laparoscopic surgery is done?

  • Information on cholelithiasis was not available in our study data. As explained above, our findings are supporting the hypothesis that common hormonal or metabolic causes underneath both cholecystectomy and gynecological cancers and not the surgical removal of gallbladder per se. However, we will check these interesting ideas in further studies.

  1. The risk of ovarian cancer increased by 35% just in a small 6 months window following the surgery. I think that the interpretation that you proposed, namely a possible pain-based misdiagnosis, is relevant enough to be included in the abstract, highlighting that it is worth doing efforts to exclude ovarian cancer before cholecystectomy.

  • This sentence was added to the conclusion part (last paragraph) of the Abstract and Discussion sections:
    • The risk of ovarian cancer increased by 35% just in a short period of time (6 months) following the surgery. Therefore, it is worth to rule out ovarian cancer before cholecystectomy.

  1. Other works have explored a link among cholelithiasis or cholecystectomy and colon cancer, which is also very interesting. I wonder if you have explored or will explore this link in your large complete dataset of Sweden patients.

  • Yes, we are also working on the topic of gastrointestinal cancers (including colon cancer) after cholecystectomy and will publish our results soon.

Reviewer 2 Report

In this manuscript, Kharazmi E et al, assessed the risk between incidence of different gynecological cancers like breast cancer, ovarian cancer, cervical cancer, and endometrial cancer in women who underwent cholecystectomy due to gallstones. From the data, the authors identified that risk of ovarian cancer increased 6 months post-surgery. They did not find such a correlation in the case of cervical cancer. However, increased risks for breast cancer and endometrial cancers were indeed identified. The study has its strength in the large data set used for the analysis and identification of such risk factors will help in early identification of patients who could potentially develop these cancers. There are few concerns with the study that are enlisted below:

  • Did the authors check for any correlations between surgical gall bladder removal and cancer stage/ disease progression and cancer sub-types e.g., triple negative breast cancer, high grade serous ovarian carcinoma, clear cell carcinoma, etc? Were there any differences in the survival or therapeutic response of the patients who had undergone cholecystectomy?
  • It would be interesting to know if the family history of cancer would be a confounding factor for the increased risk of these patients developing cancer despite undergoing cholecystectomy. Was this considered as part of the data analysis?
  • In the discussion section, the authors state that there was a 4-8% increased risk of breast cancer, but from a clinical standpoint how significant is that? What proportion of women undergo cholecystectomy in a year and how many would develop breast cancer among them? It would be interesting to know the statistics that would give a better appreciation of the risk percentages identified.
  • Minor comment: Please check the formatting of table 1

Author Response

  • Did the authors check for any correlations between surgical gall bladder removal and cancer stage/ disease progression and cancer sub-types e.g., triple negative breast cancer, high grade serous ovarian carcinoma, clear cell carcinoma, etc? Were there any differences in the survival or therapeutic response of the patients who had undergone cholecystectomy?
    • Thank you very much for your comment. Information on subtypes of cancers (e.g., triple negative breast cancer) and therapeutic response of patients were not available on our study data. However, as these are very interesting topics for further research in this field, we added following sentences to the Discussion section as a direction for further studies (page 7, paragraph 5):
      • Information on subtypes of cancers (e.g., triple negative breast cancer) and therapeutic response of patients were not available on our study data. Further studies are needed to find the association between surgical gallbladder removal and cancer stage, disease progression and cancer sub-types e.g., triple negative breast cancer, high grade serous ovarian carcinoma, clear cell carcinoma, etc. It is also warranted to find any differences in the survival or therapeutic response of the patients who had undergone cholecystectomy.

  • It would be interesting to know if the family history of cancer would be a confounding factor for the increased risk of these patients developing cancer despite undergoing cholecystectomy. Was this considered as part of the data analysis?
    • To our knowledge, there is no report about this to consider family history as a confounder. However, as this is a very interesting topic for further research on this field, we added following sentences to the Discussion section as a direction for further studies (page 7, paragraph 5):
      • It would be worth to investigate in further studies whether such associations are significantly different in the presence or absence of family history of the corresponding gynecological cancer.

  • In the discussion section, the authors state that there was a 4-8% increased risk of breast cancer, but from a clinical standpoint how significant is that? What proportion of women undergo cholecystectomy in a year and how many would develop breast cancer among them? It would be interesting to know the statistics that would give a better appreciation of the risk percentages identified.
    • Following sentence was added to the Results section, at the end of first paragraph in page 4:
      • Proportion of cancers that theoretically could potentially attributable to cholecystectomy in the general population was at most 0.2% for breast cancer and 1% for endometrial cancer.

  • Minor comment: Please check the formatting of table 1
    • Format of Table 1 was fixed.

Reviewer 3 Report

The aim of this paper is to provide the cohort nationwide study of risk of gynecological cancers for cholecystectomized women. The review is generally well written and scientifically sound. I would suggest to provide better and more uniform formatting of References and probably slightly increase size of the Discussion section (for example, provide a description of dropping out last points on graphs of Incidence rates vs age).

Author Response

The aim of this paper is to provide the cohort nationwide study of risk of gynecological cancers for cholecystectomized women. The review is generally well written and scientifically sound. I would suggest to provide better and more uniform formatting of References and probably slightly increase size of the Discussion section (for example, provide a description of dropping out last points on graphs of Incidence rates vs age).

  • Thank you for your comment We took care of formatting of References using EndNote; however, the editorial office will modify it again according to the journal style. The size of Discussion section was increased after applying all the comments from all reviewers.
  • Following description on dropping out last points on graphs of incidence rates versus time since cholecystectomy was added to page 8, paragraph 1:
    • There were some random fluctuations at the end of the incidence curves, which were mainly due to smaller sample size at elderly ages (25-30 years after cholecystectomy).

Reviewer 4 Report

The authors examined the risk of gynecological cancers in chelocystectomied women using Swedish Cancer, Population, Patient, and Death registries. The authors found that gallbladder removal was associated with an increased risk of ovarian breast, ovarian, and endometrial cancer. This manuscript is well written and well organized. My minor comments are listed below.

Abstract

Please do not use abbreviations without definition (including tables). Once an abbreviation is defined in the main text, thereafter can it only be used throughout the manuscript.

Discussion

The authors found that there is an increased risk of ovarian cancer found in the first six months after cholecystectomy. A recent study examined the effect of opportunistic salpingectomy at the time of laparoscopic cholecystectomy for ovarian cancer prevention (PMID: 35129467). I think the results of the current study are interesting to discuss this matter.

Since the study period is wide, the authors need to discuss the effect of the year of inspection on the detection rate of breast, endometrial, ovarian, and cervical cancer. In recent years, numerous studies have reported that the detection rate of these cancers, except ovarian cancer, has been improving. Please add this point as a limitation of this study.

Conclusion

The authors mention that “A lower risk of cervical cancer was found in women who underwent cholecystectomy after the age of 50 years; this association should be validated in future studies and, if it proves true, the reasons for the negative association should be examined in detail.”

The reviewer thinks the authors may remove these statements from the conclusion of this study.

Author Response

The authors examined the risk of gynecological cancers in chelocystectomied women using Swedish Cancer, Population, Patient, and Death registries. The authors found that gallbladder removal was associated with an increased risk of ovarian breast, ovarian, and endometrial cancer. This manuscript is well written and well organized. My minor comments are listed below.

Abstract

Please do not use abbreviations without definition (including tables). Once an abbreviation is defined in the main text, thereafter can it only be used throughout the manuscript.

  • Thank you very much for your comments. Two abbreviations with displaced definitions were fixed; one in Results part of the Abstract (CI, page 1) and one in the methods section (ICD, page 2, paragraph 4). We made sure that all abbreviations used in the tables are defined in the footnotes of the tables.

Discussion

The authors found that there is an increased risk of ovarian cancer found in the first six months after cholecystectomy. A recent study examined the effect of opportunistic salpingectomy at the time of laparoscopic cholecystectomy for ovarian cancer prevention (PMID: 35129467). I think the results of the current study are interesting to discuss this matter.

  • Thank you for introducing this very interesting paper. We added following sentence about it in the Discussion (page 7, first paragraph):
    • A modeling study has also recently found that the opportunistic salpingectomy performed at the time of elective laparoscopic cholecystectomy may be a cost-effective strategy to prevent ovarian cancer among average-risk women.1

Since the study period is wide, the authors need to discuss the effect of the year of inspection on the detection rate of breast, endometrial, ovarian, and cervical cancer. In recent years, numerous studies have reported that the detection rate of these cancers, except ovarian cancer, has been improving. Please add this point as a limitation of this study.

  • The following sentence was added to the limitations of the study in the Discussion section, page 7, last paragraph:
    • In recent years, numerous studies have reported that the detection rate of gynecological cancers, except ovarian cancer, has been improving, especially of those with existing recommended screening modalities. Therefore, in addition to aging, the effect of the year of inspection on the detection rate of breast, endometrial, ovarian, and cervical cancer could have partially contributed to the overall increasing trend of incidence of these cancers by time after cholecystectomy in both cholecystectomized and reference groups. However, without using such a long-standing data, the long-term follow-up of cholecystectomy would not be possible.

Conclusion

The authors mention that “A lower risk of cervical cancer was found in women who underwent cholecystectomy after the age of 50 years; this association should be validated in future studies and, if it proves true, the reasons for the negative association should be examined in detail.”

 The reviewer thinks the authors may remove these statements from the conclusion of this study.

  • The sentence was removed from the Conclusion section.

Reference in answer letter:

1. Matsuo K, Chen L, Matsuzaki S, et al. Opportunistic Salpingectomy at the Time of Laparoscopic Cholecystectomy for Ovarian Cancer Prevention: A Cost-Effectiveness Analysis. Annals of surgery. 9000;doi:10.1097/sla.0000000000005374